Gender disparities in the association between socio-demographics and non-communicable disease risk factors among adults with disabilities in Shanghai, China

Zhang Youran 1 2
Chen Gang gchen@shmu.edu.cn 1 2
Zhang Qi 3
Lu Jun 2
Yu Huijiong 4
1 Department of Health Law and Health Inspection, School of Public Health, Fudan University , Shanghai , China
2 China Research Center on Disability Issues at Fudan University , Shanghai , China
3 School of Community and Environmental Health, Old Dominion University , Norfolk , VA , United States of America
4 Department of Rehabilitation, Shanghai Disabled Persons’ Federation , Shanghai , China
Marusic Ana
Electronic publication date: 2018 Mar 13
Publication date: 2018
Volume: 6
Electronic Location ID: e4505
Received 2017 Dec 14; Accepted 2018 Feb 24
Copyright: ©2018 Zhang et al.
Copyright year: 2018
Copyright holder: Zhang et al.
License: This is an open access article distributed under the terms of the Creative Commons Attribution License, which permits unrestricted use, distribution, reproduction and adaptation in any medium and for any purpose provided that it is properly attributed. For attribution, the original author(s), title, publication source (PeerJ) and either DOI or URL of the article must be cited.
License URL: https://creativecommons.org/licenses/by/4.0/

Keywords: Non-communicable disease risk factors, Disability, Gender disparity

Funding: Shanghai Disabled Persons’ Federation (SHDPF) 111 Project B16031 This work was supported by Shanghai Disabled Persons’ Federation (SHDPF) and the 111 Project (grant number: B16031). The funders had no role in study design, data collection and analysis, decision to publish, or preparation of the manuscript.

==============================
Background

Non-communicable disease (NCD) risk factors can co-exist with disability and cause a greater burden on the health status of adults with disabilities. A lack of egalitarian social policies in China may result in gender disparities in the NCD risk factors of adults with disabilities. However, little is known about the gender disparities in the association between socio-demographics and NCD risk factors among adults with disabilities in China; consequently, we examined this association among adults with disabilities in Shanghai, China.

Methods

We used the health examination data of 44,896 adults with disabilities in Shanghai in 2014. Descriptive analyses and logistic regression models were conducted to estimate gender disparities in the association between socio-demographics, disability characteristics, and four selected NCD risk factors among adults with disabilities—including high blood pressure, high blood glucose, high blood lipids, and being overweight. We estimated marginal effects (MEs) on NCD risk factors between gender and other confounders.

Results

Women with disabilities were about 11.6 percentage points more likely to suffer from high blood lipids and less likely to develop the other three risk factors than men were. The association of age group, residence permit, education level, marital status, and disability type with health outcomes varied by gender among adults with disabilities. The difference in age effects between men and women was more pronounced in older age groups. Urban residence was associated with less risk of high blood pressure risk among women (ΔME =  − 0.035, p < 0.01), but no significant difference in other NCD risk factors. Education remained a major protective factor against high blood pressure, high blood glucose and being overweight among women with disabilities (MEs < 0, p < 0.05); however, this did not hold for men. The difference in marriage effects between men and women was observed in high blood lipids (ΔME =  − 0.048 for the married group and −0.054 for the divorced or widowed group) and overweight individuals (ΔME =  − 0.091 for the married group and −0.114 for the divorced or widowed group). Women with intellectual disabilities or mental disabilities reported worse health conditions than men did.

Discussion

Preventive strategies and interventions on NCD risk factors for adults with disabilities should take into account gender disparities in these socio-demographic effects. Rural women or poorly educated women with disabilities can be a vulnerable population that requires more health education and promotion strategies. Health education for caregivers of women with intellectual or mental disabilities may also play a vital role in preventing their NCD risk factors.

Introduction

More than one billion people in the world live with some form of disability, accounting for 15.6% of the total population (WHO, 2011). The average prevalence rates of women/men with disabilities are 19.2% and 12.0% respectively, ranging from 14.4% and 9.1% in developed countries to 22.1% and 13.8% in lower income countries (WHO, 2011). Among them, a population of more than 85 million individuals with disabilities live in China (China Disabled Persons’ Federation, 2007; The People’s Republic of China, 2016) and nearly half (48.45%) are women.

Disabilities, as well as lack of egalitarian social policies and accommodation, severely impact individuals’ lives in a variety of ways. Individuals with disabilities are among the most marginalized groups in society, facing high rates of unemployment, poverty and more health challenges (WHO, 2011).

Non-communicable disease (NCD) risk factors, such as uncontrolled high blood pressure, impaired fasting glucose, hyperlipidemia, and being overweight, can co-exist with disability and add to the burden on the health status of these populations (WHO, 2013). These health risks could potentially decrease the balance, strength, endurance, fitness, and flexibility of these individuals and increase the risk of spasticity, depression, and other health problems (Rimmer, 1999). When these risk factors further develop into NCD, they can significantly increase mortality and health care expenditure with respect to people with disabilities.

Recent literature has revealed that women have a higher disability rate than men do (Subramaniam et al., 2013; Tas et al., 2011; Bora & Saikia, 2015). Although women with disabilities have a longer life expectancy, they tend to suffer more from non-life-threatening conditions. It is well established that there are gender disparities with respect to the risk of developing a disability or the contribution of chronic conditions to the burden of the disability (Whitson et al., 2010; Yokota et al., 2016). Moreover, some researchers have found that gender differences exist in the prevalence of risk factors among people with disabilities (Krause & Broderick, 2004; Hinkle, Smith & Revere, 2006; McDermott et al., 2007). There are also some researchers who have examined the effect of socio-demographic and disability characteristics on NCD risk factors in this population (Chen et al., 2011; Lin et al., 2013; Kang et al., 2016; Horner-Johnson et al., 2013). These studies helped build the evidence base for exploring health disparities among them. However, few researchers have sufficiently investigated gender differences in the association between socio-demographics, disability characteristics, and NCD risk factors among people with disabilities, especially in China. To prevent NCD risk factors effectively, Chinese health sectors need to tailor suitable measures for men and women with disabilities respectively. Uncovering how socio-demographic and disability characteristics influence the prevalence of these NCD health risks in men and women with disabilities would be conducive to deploying more accurate interventions.

To examine the gender disparities in a population with disabilities, we used the health examination data of 44,896 adults with disabilities in Shanghai, China. The purpose was to: (1) compare population-level estimates of the prevalence of various health risks between men and women with disabilities; and (2) compare the marginal effects (MEs) of socio-demographic and disability characteristics on NCD risk factors between men and women with disabilities.

Materials and Methods

Data source and ethics statement

Since 2004, Shanghai has provided free yearly health examination services for people with disabilities. Two municipal rehabilitation centers provide professional and comprehensive health checks for about 40,000 people with disabilities every year. In total 46,108 people with disabilities (accounting for almost 11.49% of total individuals with disabilities in Shanghai) accessed health examination services at the two municipal rehabilitation centers from January 1, 2014 to December 31, 2014. After eliminating the data of persons aged younger than 18 years (n = 66) or those with missing data on identified variables (n = 1,146), 44,896 persons with disabilities were included in analyses.

The health examination records of these individuals and their socio-demographics were collected by the Shanghai Disabled Persons’ Rehabilitation Comprehensive Information Platform (SHDPRCIP), which was established by the Shanghai Disabled Persons’ Federation, during this period. The institutional review board (IRB) of the Fudan University School of Public Health (IRB #2015-08-0563) authorized this study.

We selected four important NCD risk factors: high blood pressure, high blood glucose, high blood lipids, and being overweight, which all exert considerable influence on public health. Only adults aged over 18 years were included in our study. Records with missing data on the selected variables were excluded.

Dependent variables

Persons with systolic blood pressure ≥140 mmHg and/or diastolic pressure >90 mmHg were denoted as having high blood pressure (Revised Commission on Chinese Prevention and Treatment Guideline for Hypertension, 2011). A fasting blood glucose level ≥6.1 mmol/L was regarded as high based on Chinese Prevention and Treatment Guideline for Type 2 Diabetes (2013) (Chinese Diabetes Federation, 2014). Total cholesterol levels ≥5.2 mmol/L or triglyceride levels ≥1.7 mmol/L were classified as high blood lipids according to Chinese Adults’ Prevention and Treatment Guidelines for Dyslipidemia (2016) (Revised Joint Commission on Chinese Adults’ Prevention and Treatment Guidelines for Dyslipidemia, 2016). Moreover, a body mass index ≥24 kg/m2 was considered as overweight based on the guidelines recommended for the Chinese population (Zhou, 2002). All dependent variables were categorized into binary outcomes.

Conceptual model and covariates

The main explanatory variable was gender. Two groups of factors were also controlled: socio-demographics (age group, residence permit, education level, and marital status) and disability characteristics (disability type and disability severity). Residence permit in China contains only two types: urban and rural. Many social security welfare systems, such as health, housing, education and pensions, are based on the household registration system. Urban residence may receive more benefits than rural residence. Age group was defined as 18–29, 30–39, 40–49, 50–59, 60–69, and 70 or older. The education level groups were elementary school or lower, middle school, high school, and college or higher. Marital status was classified as never married, married, and divorced or widowed.

Based on the Classification and Grading Criteria of Disability (GB/T 26341-2010) (Standardization Administration of the People’s Republic of China, 2011), specialized medical institutions designated by the Shanghai Disabled Persons’ Rehabilitation performed the disability evaluation. The evaluation outcomes (i.e., disability type and disability severity) were registered in the SHDPRCIP. Disability type included hearing disability, speech disability, visual disability, physical disability, intellectual disability, mental disability, and multiple disabilities. Those with hearing disability or speech disability were grouped together (Zheng et al., 2011). Disability severity was classified into four levels by related function scores of every disability type according to standard Chinese criteria (Standardization Administration of the People’s Republic of China, 2011). Level 1 was the most serious disability status and Level 4 was the mildest level. Multiple disabilities refer to two or more kinds of disabilities. Multiple disabilities are graded according to the grading criteria of the severest disability.

Statistical analysis

Socio-demographics, disability characteristics and NCD risk factors were compared between men and women using standard descriptive methods, such as frequency distribution and chi-square test. Logistic regression models were used to test the associations of the risk factors with socio-demographics and disability factors (model 1: baseline model). Model 2, an interaction model, additionally included the interactions between gender and other covariates. To clearly present the different effects between the two genders, MEs, instead of odds ratios, were estimated in both models, which indicated, on average, that women had more or less probabilities to have these NCD risk factors than men. MEs of socio-demographics and disability factors were estimated separately in men and women to compare the different socio-demographic effect on NCD risk factors between men and women with disabilities.

Four multiple logistic regressions (one for each NCD risk factor) were performed in each model. Stata version 12 was used for all calculations. Statistical significance was considered as p < 0.05.

Table 1 Socio-demographic and disability characteristics of the sample according to gender.

Characteristics	Total	Men	Women	p	
	n	%	n	%	n	%		
Age group								
18–29	1,406	3.13	768	3.23	638	3.02	<0.01	
30–39	2,740	6.10	1,376	5.79	1,364	6.46	
40–49	5,121	11.41	2,483	10.44	2,638	12.50	
50–59	17,185	38.28	8,942	37.60	8,243	39.05	
60–69	14,875	33.13	8,184	34.41	6,691	31.69	
70 +	3,569	7.95	2,032	8.54	1,537	7.28	
Residence permit								
Rural	9,271	20.65	4,883	20.53	4,388	20.79	0.50	
Urban	35,625	79.35	18,902	79.47	16,723	79.21	
Education level								
Elementary school or lower	10,737	23.92	4,950	20.81	5,787	27.41	<0.01	
Middle school	22,179	49.40	12,083	50.80	10,096	47.82	
High school	9,985	22.24	5,423	22.80	4,562	21.61	
College or higher	1,995	4.44	1,329	5.59	666	3.15	
Marital status								
Never married	5,834	12.99	4,021	16.91	1,813	8.59	<0.01	
Married	36,137	80.49	18,430	77.49	17,707	83.88	
Divorced or widowed	2,925	6.52	1,334	5.61	1,591	7.54	
Disability type								
Hearing or speech	4,417	9.84	2,320	9.75	2,097	9.93	<0.01	
Visual	10,496	23.38	4,960	20.85	5,536	26.22	
Physical	22,443	49.99	12,625	53.08	9,818	46.51	
Intellectual	5,013	11.17	2,639	11.10	2,374	11.25	
Mental	2,019	4.50	951	4.00	1,068	5.06	
Multiple	508	1.13	290	1.22	218	1.03	
Disability severity								
Level 1	3,583	7.98	1,943	8.17	1,640	7.77	<0.01	
Level 2	5,865	13.06	3,283	13.80	2,582	12.23	
Level 3	13,027	29.02	7,362	30.95	5,665	26.83	
Level 4	22,421	49.94	11,197	47.08	11,224	53.17	

Results

Table 1 presents participants’ socio-demographic and disability characteristics according to gender. For the overall sample, the average age (±SD) was 56.18 ±11.11 years and 52.98% were men. Most people were aged 50–59 years (37.60% men vs. 39.05% women). About 80% of the sample were aged older than 50 years. Men had received more education than women had, but were less frequently married. The proportion of men (53.08%) with physical disabilities was higher than it was for women (46.51%); however, the proportion of men with visual disabilities was lower (20.85% in men vs. 26.22% in women). Compared to women, men were diagnosed as having more severe disabilities. Overall, there were significant differences in the socio-demographic and disability characteristics between men and women, except for residence permit.

The prevalence of NCD risk factors by gender is displayed in Table 2. High blood pressure, high blood lipids, and being overweight reached more than 40% in both gender groups. High blood lipids was the most common of the four selected risk factors in people with disabilities (58.56%). High blood lipids were most common in women (64.42%), which was higher than the rate for men (53.37%). However, the proportions of the other three indicators in men were higher than those in women. All the differences were statistically significant.

Table 2 NCD risk factors according to gender.

	Total	Men	Women	p	
	n	%	n	%	n	%		
High blood pressure	19,180	42.72	10,548	44.35	8,632	40.89	<0.01	
High blood glucose	8,869	19.75	5,114	21.50	3,755	17.79	<0.01	
High blood lipids	26,293	58.56	12,693	53.37	13,600	64.42	<0.01	
Overweight	22,091	49.20	12,176	51.19	9,915	46.97	<0.01	

Figure 1 shows the MEs of women compared with men on NCD risk factors. For example, on average, women were about 11.6 percentage points more likely than men to develop high blood lipids, and women were less likely to develop the other three risk factors. The gender difference in MEs was significant but the difference between two models was non-significant.

Figure 1 MEs of women vs. men on NCD risk factors.

Table 3 presents the MEs of demographic and disability characteristics in men and women, respectively, and the gender difference in the MEs of these socio-demographic effects. A positive ΔME means the women’s ME is greater than men’s ME in values, and vice versa. The gender difference in age effects was more pronounced in older age groups. For example, the ΔME in age effects on high blood pressure was −0.024 and −0.014 for those aged 30–39 and 40–49 years, respectively (although non-significant). However, the gender differences in age effects were 0.067, 0.073, and 0.128 in older groups. In other words, age effects in women were smaller than those in men in younger age groups, but significantly larger in older age groups. Similar patterns can be observed for other NCD groups. Urban residence was associated with less risk of high blood pressure among women (ΔME =  − 0.035, p < 0.01), but no significant difference in other NCD risk factors. In other words, in women, the protective effect of urban residence was much larger and significant than in men with respect to high blood pressure. Education was a protective factor against high blood pressure, high blood glucose and being overweight among women with disabilities; i.e., all MEs were negative. More educated women had better health statuses (the absolute values of the MEs increased). However, this did not appear true for men. The ΔME in education was significantly pronounced in these three NCD groups (p < 0.05). Intuitively speaking, education had a significantly protective effect in women, but not in men. The ΔME in marriage effects were observed in high blood lipids (ΔME =  − 0.048 for the married group and −0.054 for the divorced or widowed group, p < 0.05) and being overweight (ΔME =  − 0.091 for the married group and −0.114 for the divorced or widowed group, p < 0.01). Regarding disability type effects, gender differences was found in mental disability group and intellectual disability group. The gender difference in the mental disability effect was positive on high blood pressure (ΔME = 0.083, p < 0.01), high blood glucose (ΔME = 0.073, p < 0.01), and being overweight (ΔME = 0.125, p < 0.01), but no significant difference in high blood lipids. Women with intellectual disability were more vulnerable to high blood lipids (ΔME = 0.069, p < 0.01) and becoming overweight (ΔME = 0.081, p < 0.01). Disability severity had a trivial effect on these health risks in both men and women, and the gender differences in disability severity were non-significant.

Table 3 MEs of demographic and disability characteristics between men and women with disabilities in interaction model.

	High blood pressure	High blood glucose	High blood lipids	Overweight	
	Men	Women	ΔME	Men	Women	ΔME	Men	Women	ΔME	Men	Women	ΔME	
Age group													
18–29	Reference												
30–39	0.059**	0.035*	−0.024	0.061**	0.029**	−0.031*	0.088**	0.047*	−0.041	0.014	−0.029	−0.043	
40–49	0.132**	0.119**	−0.014	0.115**	0.068**	−0.048**	0.149**	0.155**	0.006	−0.007	0.006	0.013	
50–59	0.216**	0.283**	0.067**	0.180**	0.137**	−0.044**	0.142**	0.369**	0.226**	−0.051*	0.043	0.094**	
60–69	0.312**	0.385**	0.073**	0.197**	0.163**	−0.035*	0.106**	0.418**	0.312**	−0.037	0.081**	0.118**	
70 +	0.420**	0.548**	0.128**	0.214**	0.204**	−0.009	0.029	0.406**	0.377**	−0.052*	0.126**	0.178**	
Residence permit													
Rural	Reference												
Urban	−0.008	−0.043**	−0.035**	0.012	−0.001	−0.013	0.013	0.021*	0.008	0.036**	0.023**	−0.013	
Education level													
Elementary school or lower	Reference												
Middle school	0.011	−0.032**	−0.043**	0.005	−0.022**	−0.027**	0.015	0.004	−0.011	0.000	−0.069**	−0.068**	
High school	0.025*	−0.067**	−0.092**	0.007	−0.039**	−0.045**	0.024*	0.027**	0.003	0.008	−0.115**	−0.122**	
College or higher	0.009	−0.097**	−0.106**	−0.036**	−0.066**	−0.030	0.030	0.029	−0.001	0.014	−0.140**	−0.154**	
Marital status													
Never married	Reference												
Married	−0.031**	−0.029	0.002	−0.020*	0.000	0.020	0.014	−0.034*	−0.048**	0.092**	0.001	−0.091**	
Divorced or widowed	−0.033*	−0.019	0.014	−0.024	−0.006	0.018	0.016	−0.038*	−0.054*	0.081**	−0.032	−0.114**	
Disability type													
Hearing or speech	Reference												
Visual	0.010	0.010	0.000	0.030**	0.004	−0.026	0.013	0.018	0.005	0.013	0.022	0.008	
Physical	0.019	0.000	−0.019	0.013	−0.009	−0.022	0.019	0.017	−0.002	0.045**	0.065**	0.021	
Intellectual	0.005	0.017	0.012	0.013	0.045**	0.032	−0.059**	0.010	0.069**	0.023	0.104**	0.081**	
Mental	−0.092**	−0.009	0.083**	0.025	0.098**	0.073**	0.046*	0.044*	−0.003	0.108**	0.233**	0.125**	
Multiple	0.008	−0.078*	−0.086	−0.018	0.019	0.037	0.023	0.043	0.019	−0.004	−0.013	−0.009	
Disability severity													
Level 1	Reference												
Level 2	0.036*	−0.019	−0.055**	0.004	0.007	0.004	−0.026	−0.021	0.005	0.009	−0.005	−0.014	
Level 3	−0.016	0.009	0.024	−0.027*	−0.014	0.013	−0.021	−0.026	−0.004	−0.015	0.003	0.017	
Level 4	0.008	0.019	0.011	−0.005	0.005	0.010	−0.028*	−0.024	0.004	0.003	0.010	0.007	
Notes.

* p < 0.05 with logistic regression.

** p < 0.01 with logistic regression.

Discussion

In Shanghai, women with disabilities tended to be younger and less educated and were more likely to be married than men were. The prevalence rates of high blood pressure, high blood glucose, and being overweight were higher in men with disabilities than in women; however, this was not so for high blood lipids. Our study disclosed the complex gender disparities in socio-demographic effects on NCD risk factors. Previous literature only used gender as simple control variable in the analyses regarding NCD risk factors in adults with disabilities. However, this study suggests the gender effect can vary by different age, education, or disability types.

For example, the changing signs of ΔME across age groups indicated that age generates a strong effect in men on NCD risk factors; however, in older populations, age effects were larger in women, which suggests we may need to pay more attention to chronic illness in elderly women than in elderly men with disabilities (Hosseinpoor et al., 2012).

The gender disparity in an urban protective effect is also worth discussion. Since Shanghai has a high degree of urbanization and has a greatly reduced rural–urban gap in health care access (Shanghai Municipal Statistics Bureau, 2016; Center for Health Statistics and Information, 2013), no significant difference regarding NCD risk factors was observed in men with disabilities. However, rural women can still be a target intervention group to prevent or reduce chronic illness (Michele et al., 2010; National Health and Family Planning Commission of the People’s Republic of China, 2011; Gaziano et al., 2008; Ibrahim & Damasceno, 2012).

Education remained a major protective factor against these selected risk factors among women with disabilities; although, people with disabilities were less educated than the general population in China (China Disabled Persons’ Federation, 2013). Interestingly, higher education does not play a positive role for men with disabilities. The protective effect was particularly important for women with disabilities. As a result of the segregated labor markets for people with disabilities, women who are more educated may have more opportunities to get jobs, earn more money, and obtain healthier food, and they may also be more likely to take advantage of comprehensive rehabilitation policies and health care utilities (Piao et al., 2015; Lancet, 2011). Men with disabilities of different educational backgrounds may not differ regarding these circumstances above (Zajacova & Montez, 2017). Poorly educated women with disabilities can be a vulnerable population that needs more health education and promotion strategies.

Results from our analysis showed that marital status was significantly negatively associated with developing high blood pressure and high blood glucose and positively associated with gaining body mass in men. These findings echo the literature demonstrating the association between longevity and better health status with married status, particularly for men (August & Sorkin, 2010; Wilson, 2012). However, gender disparities in the association between marriage and the selected health outcomes appeared in high blood lipids and being overweight. Although women with disabilities were likely to have health benefits through marriage, they may also suffer from the effects of caregiver burden since they often devote themselves to caring for their family members, especially partners with disabilities. Moreover, being a wife with disabilities could make women feel stressed and find their role restrictive and frustrating, which could lead to unintentional weight loss (Trevisan et al., 2016). All these may explain the lower likelihood of developing high blood lipids and becoming overweight in women than in men.

In general, people with mental disabilities faced worse health outcomes in comparison with those with hearing or speech disabilities. A credible explanation for the high incidence of NCD risk factors in the mental disability group may be less outdoor activity because of their difficulties with communication and social integration. Regarding the gender disparities, women with intellectual disabilities or mental disabilities reported worse health conditions than men did. This is alarming because women may have more sensitive and fragile psychological characteristics, leading to a more closed living environment and worse life state. More aids and supports are needed to help them integrate into the community and adapt to healthy lifestyle (Merrick, Kandel & Morad, 2003). Since women with intellectual disabilities or mental disabilities lack independent decision-making ability, including the capacity for health consciousness, health education for their caregivers can play a vital role in preventing their NCD risk factors.

Limitations

This study had some limitations. First, our study was cross-sectional; therefore, causality cannot be inferred. Since disability and the health risks affect each other over time, it is critical to consider mutual influences to obtain a more accurate representation of gender differences when examining disability. Second, this study did not control for many societal factors, such as the cause of disability, years of disability, employment status, health insurance coverage, individual income, and some behavioral factors, including taking medications for their conditions, physical activity level (Sahlin & Lexell, 2015; Carroll et al., 2014). It is critical for future studies to consider more covariates to determine a more complete picture of gender differences. Finally, this study collected secondary health examination data from the SHDPRCIP. Some people with disabilities, for instance those with severe disabilities or those who received an occupational health examination, may have been unwilling to participate in the initial health examination. Potential selection bias could exist in the sample. Nevertheless, these concerns, to some extent, were mitigated by the large sample size and objective indicators of individuals with disabilities.

Conclusions

This study highlighted the need for targeted public policies and actions on NCD risk factor prevention and management between men and women with disabilities living in Shanghai. When policymakers target different socio-demographic groups for NCD prevention and intervention, they need to adopt gender-specific policies to ensure a balanced approach to promote general health in men and women.

Supplemental Information

Data S1 Raw data

Click here for additional data file.

We would like to thank Qi Kang, Feixia Liu, Qian Wang, Lian Zong, and Xiaolan Liu for advice.

Additional Information and Declarations

Competing Interests

Author Contributions

Ethics

Data Availability

The authors declare there are no competing interests.

Youran Zhang conceived and designed the experiments, performed the experiments, analyzed the data, contributed reagents/materials/analysis tools, prepared figures and/or tables, authored or reviewed drafts of the paper, approved the final draft.

Gang Chen conceived and designed the experiments, authored or reviewed drafts of the paper, approved the final draft.

Qi Zhang conceived and designed the experiments, contributed reagents/materials/analysis tools, authored or reviewed drafts of the paper, approved the final draft.

Jun Lu and Huijiong Yu conceived and designed the experiments, authored or reviewed drafts of the paper, approved the final draft, conceived and organized the survey.

The following information was supplied relating to ethical approvals (i.e., approving body and any reference numbers):

The institutional review board (IRB) of the Fudan University School of Public Health (IRB #2015-08-0563) authorized this study.

The following information was supplied regarding data availability:

The raw data are provided in a Supplemental File.

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
