# Peer review of "Gender disparities in the association between socio-demographics and non-communicable disease risk factors among adults with disabilities in Shanghai, China"

_PeerJ, doi:10.7717/peerj.4505_

## Round 0.1 · original submission · Major Revisions

Please pay special attention to the comments of the Reviewer #1.

·

Basic reporting

The authors present a well-written study that may reveal gaps in care and preventive measures for chronic illnesses in disabled people in Shanghai, China. The strengths of the work presented includes its applicability to populations similar to that of Shanghai and the authors addressed their objectives in the analysis and discussion. Additionally, the authors presented conclusions according to the objectives and data. However, the study has several weaknesses in the Methodology that the authors should revise before re-submitting as this work does fall within the aims and scope of the journal. In particular, please find below major and minor comments and suggestions according to the section of the manuscript, thank you. Further, please find additional comments in the manuscript marked with either comments or the Track Changes tool in Word.

Experimental design

Major comments
Methods
Data source
1. In the first paragraph - regarding the health checks and health examination records by the Shanghai Disabled Persons' Rehabilitation Comprehensive Information Platform, could the authors please provide additional information about who performed the disability diagnosis and confirmation back in 2014? For example, professional psychiatrists?

Dependent variables
1. In the first paragraph – concerning the criteria for defining cut offs – would the authors consider using Chinese national guidelines or WHO guidelines to define cutoffs for high blood glucose, cholesterol, and triglycerides?

Covariates
1. If the SHDPRCIP has collected comprehensive information from participants, then it would be crucial to include (or explain data unavailability) several additional other covariates: those taking medications for their conditions and physical activity level (decreased physical activity is associated with diabetes and cardiovascular disease (1, 2)).
2. In the second paragraph of this section the authors describe the classification of disability severity. In order to reduce any bias in the classification of the disabilities into levels, which authors (Y.Z., G.C., etc.) performed the ratings? Were the ratings performed independently and disagreements resolved through consensus discussion?
3. How did the authors handle the data of individuals with comorbid disabilities? As these data already existed from the Shanghai Disabled Persons' Rehabilitation Comprehensive Information Platform (SHDPRCIP) collected back in 2014 and your dataset only includes one disability per individual, could the authors please note that either the SHDPRCIP did not collect information on more than one disability or for some other reason only one disability per individual is accounted for? It would be extremely misleading to readers and lead to bias in the interpretation of the association between sociodemographic factors and NCDs in the adults in this study if individuals with more than one disability are excluded from the current study's models. Please state clearly why comormid conditions were not collected or why you excluded them from your models. This important aspect is missing from the current report.

Validity of the findings

Results
1. Major comments regarding covariates used in the analysis are in section 2.
2. Table 1 – please define hukou in the table or restrict usage of the word to the main text.

Discussion
Limitations
1. In the first paragraph, fourth sentence - the authors mention that they did not control for 'health insurance', but health insurance coverage still does not mean that a person can access care or use healthcare resources. Could the authors please briefly mention this distinction in this section?

Minor comments
Please change 'lipid' to 'lipids' throughout the manuscript.
In the Introduction section, second paragraph, line 64 – please delete 'the' before the word society, line 65 – please delete 'the' before the word spreading.
In line 93 – please change 'the' to 'a' population with disabilities…
In line 117 – please describe your instituion as 'the Fudan University School of Public Health.'
Line 121 – please provide a reference
Line 128 – please delete 'and'. The word 'or' is sufficient enough.

Results

1. Table 3 – please change footnotes in the table to *p < 0.05 with logistic regression; **p < 0.01 with logistic regression, for example. Please delete the word 'means' before the p value.

References
1. Sahlin KB, Lexell J. Impact of Organized Sports on Activity, Participation, and Quality of Life in People With Neurologic Disabilities. PM & R : the journal of injury, function, and rehabilitation. 2015;7(10):1081-8. Epub 2015/04/02.
2. Carroll DD, Courtney-Long EA, Stevens AC, Sloan ML, Lullo C, Visser SN, et al. Vital signs: disability and physical activity--United States, 2009-2012. MMWR Morbidity and mortality weekly report. 2014;63(18):407-13. Epub 2014/05/09.

Reviewer 2 ·

Basic reporting

Thank you for inviting me to review this interesting manuscript. The reporting in the manuscript is in line with the PeerJ requirements. The abstract is structured and includes appropriate subheadings as well as the manuscript itself. The literature is appropriately referenced and the background provides sufficient context. The manuscript includes labelled figures and tables. My suggestion would be to use clearer language in some places. For example, it is unclear why past tense is used in 2nd and 3rd sentence of the manuscript. The sentence, row 54-56 is unnecessarily long and vague etc. "Pertinently" in line 75 may not be the most suitable descriptor. I would also like to suggest clearer presentation of results across the manuscript, including the abstract. There is a lot of information and in order to reinforce the message, I would like to suggest greater consistency and clarity in reporting of findings. E.g. "women had a significantly larger negative urban hukoku effect on high blood pressure" is unclear. Does that mean that urban residence was more or less likely associated with high blood pressure in women?

Experimental design

The objectives of the study are clearly defined and meaningful. I would suggest clearly referencing similar and pertinent research, e.g. on disability prevalence or NCD risk factors prevalence in China as well as any similar studies from other settings. While I do not have statistical background and am unfamiliar with marginal effect approach to presentation of data, the analyses seem relevant and appropriate and described with sufficient detail.

Validity of the findings

It is important to present the study findings in the light of existing evidence. Presented data seems robust, statistically sound and controlled. Conclusions are clear and relate to the study aims.

Additional comments

This is an interesting manuscript which provides important insight into the areas and population groups which should be targeted with future public health policies. I would like to suggested greater clarity in presentation of findings both in abstract as well as other parts of the paper. There is a need for some editing throughout the manuscript.

---

## Round 0.2 · Minor Revisions

Thank you for considering the reviewers' comments. There are still some small outstanding issues that need to be addressed.

·

Basic reporting

Thank you for addressing my comments and suggestions as well as those from Reviewer #2. The current manuscript reflects changes that have improved the reporting of your data sources and analyses. Before re-submission, however, please note some additional minor suggestions for revision below.

Abstract
p. 1, line 35 - Women with…percentage 'points' please add an 's' to the word 'point.'

Experimental design

Minor comments

Materials and Methods
Data Source
p. 3, line 117 – please delete 'ed' from the word 'exerted.'
p. 3, line 124 – As the authors already mentioned blood glucose in line 123, please delete ' blood glucose' after the word 'high.'

Conceptual Model and Covariates
p.3, line 143 – please add 'the' before Classification and Grading Criteria of Disability.
p. 3, line 146 – please add 'the' before SHDPRCIP.

Statistical Analysis
p. 4, line 171-172 – could you please revise the last sentence to '… was considered at p < 0.05' instead of 'taken as' and without the word 'value.'

Validity of the findings

Minor comments

Results
p.4, line 197 – The authors' findings are meaningful, so please change the word 'insignificant (unimportant)' to non-significant (lacking statistical significance).'
p. 4, line 199 – please surround the word 'respectively' with commas: …women, respectively, …
p. 4, line 200 – please add 'A' at the beginning of the sentence and the word 'that' before 'women's ME…'
p. 5, line 225 – please add 'the' before 'mental' in this sentence.
p. 5, line 232 – please change 'insignificant' to 'non-significant.'
Discussion

Limitations
p. 6, lines 294 – 297 – thank you for adding additional information to the Limitations, however, please re-word the following sentence for clarity (for example): 'Second, this study did not control for many societal factors, such as the cause of disability, years of disability, employment status, health insurance coverage, use of healthcare resources, individual income, and some behavioral factors, including medication use for their conditions and physical activity level [39-40].' Please delete the text and sentence after '…physical activity level [39-40]' so that the next sentence after the one above is: 'It is critical for future studies to consider more covariates to determine a more complete picture of gender differences.'

---

## Round 0.3 · accepted · Accept

Thank you for the final revision of your manuscript in which you addressed the comments of the reviewer.